# Evaluating novel engagement mechanisms, yields and acceptability of tuberculosis screening at retail pharmacies in Ho Chi Minh City, Viet Nam

**Phuong M. T. Tran[1], Thu A. Dam[1], Huy B. Huynh[1], Andrew J. Codlin[1]\*, Rachel J. Forse[1], Ha M. T. Dang[2], Vinh V. Truong[2], Lan H. Nguyen[2], Hoa B. Nguyen[3], Nhung V. Nguyen[3], Jacob Creswell[4], Farouk Meralli[5], Fukushi Morishita[6], Thuy T. T. Dong[1], Giang H. Nguyen[7], Luan N. Q. Vo[1,7]**

**1** Friends for International TB Relief, Ha Noi, Viet Nam, **2** Pham Ngoc Thach Hospital, Ho Chi Minh City, Viet Nam, **3** National Lung Hospital, Ha Noi, Viet Nam, **4** Stop TB Partnership, Geneva, Switzerland, **5** mClinica, Singapore, Singapore, **6** World Health Organization Regional Office for the Western Pacific, Manila, Philippines, **7** IRD VN, Ho Chi Minh City, Viet Nam

\* andrew.codlin@tbhelp.org

## Abstract

Pharmacies represent a key health system entry point for people with TB in Viet Nam, but high fragmentation hinders their broader engagement. Professional networking apps may be able to facilitate pharmacy engagement for systematic TB screening and referral. Between September and December 2019, we piloted the use of a social networking app, SwipeRx, to recruit pharmacists for a TB referral scheme across four districts of Ho Chi Minh City, Viet Nam. We measured chest X-ray (CXR) referrals and TB detection yields at participating pharmacies and fielded 100 acceptability surveys, divided into pharmacists who did and did not make a CXR referral. We then fitted mixed-effect odds proportional models to explore acceptability factors that were associated with making a CXR referral. 1,816 push notifications were sent to pharmacists via the SwipeRx app and 78 indicated their interest in participating; however, only one was within the pilot's intervention area. Additional in-person outreach resulted in the recruitment of 146 pharmacists, with 54 (37.0%) making at least one CXR referral. A total of 182 pharmacy customers were referred, resulting in a total of 64 (35.2%) CXR screens and seven people being diagnosed with TB. Compared to pharmacists who did not make any CXR referrals, pharmacists making at least one CXR referral understood the pilot's objectives more clearly (aOR = 2.6, 95% CI: 1.2–5.8) and they believed that TB screening increased customer trust (aOR = 2.7, 95% CI: 1.2–5.8), benefited their business (aOR = 2.8, 95% CI: 1.3–6.2) and constituted a competitive advantage (aOR = 4.4, 95% CI: 1.9–9.9). They were also more confident in using mHealth apps (aOR = 3.1, 95 CI%: 1.4–6.8). Pharmacies can play an important role in early and increased TB case finding. It is critical to highlight the value proposition of TB referral schemes to their business during recruitment. Digital networking platforms, such as SwipeRx, can facilitate referrals for TB screening by pharmacists, but their ability to identify and recruit pharmacists

**Data Availability Statement:** The data are online and publicly available from the Dryad repository: https://doi.org/10.5061/dryad.44j0zpchj.

**Funding:** This evaluation was supported with funds from the Joint TDR/WPRO Small Grants Scheme. Additional support was provided by the Stop TB Partnership's TB REACH initiative, with funding from Global Affairs Canada and USAID. These funding agencies had no role in the design of the pilot or its implementation. Members of their administrative Secretariat's (JC and FM) participated in the writing and review of this manuscript.

**Competing interests:** The authors have declared that no competing interests exist.

requires optimization, particularly when targeting specific segments of a nation-wide digital network.

## Introduction

With an estimated 1.2 million deaths annually, tuberculosis (TB) remains a major source of avoidable mortality [1]. Improving TB treatment coverage is critical for reducing TB incidence and mortality, and as a result, it is a vital component of the World Health Organization's (WHO's) End TB Strategy [2]. In Viet Nam, TB treatment coverage stands at just 60%, suggesting that two in five persons with incident TB are not being diagnosed, treated and/or reported to the National TB Control Programme (NTP) [3].

Since the deregulation and decentralization of healthcare in 1989, Viet Nam's private healthcare sector has steadily expanded [4], with private healthcare gaining popularity among providers and patients alike [5]. A particular growth area has been private retail pharmacies. Most retail pharmacies in Viet Nam employ staff with minimal formal pharmacy training, who then work under the supervision of an off-site pharmacist. In 2019, there were approximately 62,500 officially registered pharmacies across Viet Nam [6]. In Ho Chi Minh City, the country's most populous city, this number increased from just 1,814 in 2000 [7] to over 6,300 in 2019 [8]. These pharmacies often represent the initial point of contact of health-seeking for sick persons, including for a third of health-seeking individuals with TB [9, 10]. Meanwhile, studies have found that many private pharmacies do not refer people with presumptive TB for appropriate care, and along with other client-facing barriers, this behavior can contribute to delays in TB diagnosis and treatment [11] and potentially poorer treatment outcomes [12].

Consequently, the engagement of private pharmacies can be a useful intervention for TB care and prevention [13–16]. However, medical training for pharmacy staff is often limited and there is little economic incentive for pharmacies to refer their customers to other providers, as this may lead to a decline in medication sales [17]. Thus, it is important to further explore the ways private pharmacies can be engaged at scale to meaningfully participate in the effort to detect and treat people with TB [18, 19]. As the private healthcare sector has expanded, engagement efforts have faced challenges due to the burden of reporting to the NTP and a lack of efficient ways to engage providers at scale [10]. Mass media campaigns have been used to engage both general populations and specific audiences on TB knowledge, healthcare seeking, and stigma reduction, highlighting the potential for mobile and mass media to be a valuable tool in engaging a range of stakeholders [20–22].

A digital pharmacy professional networking platform called SwipeRx (mClinica Pte. Ltd., Singapore) was used by more than 18,000 pharmacists in Viet Nam in 2019, including more than 3,000 in Ho Chi Minh City where the non-governmental organization, Friends for International TB Relief (FIT), has been implementing the Proper Care in the Private Sector (PCPS) outreach model for TB [23]. PCPS works to increase detection of TB in the private sector by raising diagnostic quality and subsidizing advanced diagnostic tools and algorithms.

We piloted a private pharmacy engagement scheme using the SwipeRx app to facilitate outreach and recruitment of pharmacies and to encourage verbal screening of customers and referral of eligible people with presumptive TB. This evaluation aimed to assess the effectiveness of our novel pharmacist engagement approach, to measure diagnostic referral and TB detection yields, and to identify factors associated with a pharmacist's active participation in the pilot.

## Methods

### Pilot setting

This pilot took place between September and December 2019 in four districts (5, 8, 10 and Go Vap) of Ho Chi Minh City, Viet Nam. These districts were selected so that the pilot was embedded within the larger PCPS project, which at the time focused on engaging doctors and hospitals for TB diagnostic referrals. Together, these districts had an estimated population of 1.6 million in 2019, and notified 2,173 people with all forms of TB, accounting for 15.7% of the city's TB burden. In addition, there were 1,189 officially registered private pharmacies and drug retailers operating in the four districts during the pilot period.

### Pharmacist engagement

To promote participation in our pilot, we collaborated with SwipeRx to engage their existing user base in Ho Chi Minh City. We sent push notifications through the SwipeRx app to active user accounts registered in Ho Chi Minh City a total of five times over a one-month period; at the time, further targeting at the district-level was not possible. We also launched a Facebook campaign encouraging pharmacists in our intervention districts to join SwipeRx and to participate in the pilot. SwipeRx users indicated their willingness to participate in the pilot using a dedicated form within the SwipeRx app. We contacted those willing to participate and arranged on-site meetings to explain the pilot's procedures and to sign participation agreements.

In addition, we obtained official lists of licensed private pharmacies from the District Health Authorities. Pharmacies from these lists were approached in person to gauge their willingness to participate and whether or not they were existing SwipeRx users. For those agreeing to participate, we helped them set up SwipeRx accounts, trained them on pilot procedures and signed participation agreements. We engaged any staff member working at the pharmacy who directly dispensed and consulted clients in the pilot, regardless of their official certification or degree. Hereafter, these individuals are referred to as pharmacists in the manuscript.

### TB screening, referral and diagnosis

Participating pharmacists were asked to verbally screen their customers for TB symptoms and to refer those reporting symptoms (i.e., cough, weight loss, fever, night sweats) for additional chest X-ray (CXR) screening at the nearest public or private radiography site participating in the PCPS program [24]. Symptom screening and referrals were tracked using a custom-built mHealth platform called ACIS (Access to Care Information System, Clinton Health Access Initiative [CHAI] & TechUp, Ha Noi, Viet Nam). This system has been described in more detail elsewhere [25]. ACIS forms were embedded within the SwipeRx app so pharmacists could screen customers using an app with which they were familiar.

People with presumptive TB received a CXR voucher which could be used to avail a VND 50,000 (USD 2.16 at an exchange rate of 31 Dec 2019 of VND 23,129 to 1 USD) subsidy at participating radiography sites. The actual cost of a CXR at participating radiology sites ranged from VND 80,000 (USD 3.46) to VND 120,000 (USD 5.19), with the presumptive TB patient paying the difference. When a CXR was abnormal, people with presumptive TB were offered an Xpert MTB/RIF diagnostic test (Xpert) free of cost. People who were diagnosed with TB were actively linked to treatment at public-sector sites or with private doctors participating in the PCPS program. No additional clinical evaluations were conducted for people with abnormal CXR results, but negative Xpert results. Pharmacists received VND 10,000 (USD 0.43) for

each CXR referral recorded in ACIS, VND 40,000 (USD 1.73) for each Xpert test result and VND 500,000 (USD 21.62) for each person started on TB treatment.

## Acceptability survey

We developed an English language acceptability survey consisting of 27 questions (S1 Survey) using a theoretical framework for healthcare intervention acceptability which encompasses seven key constructs: 1) ethicality; 2) intervention coherence; 3) burden; 4) opportunity cost; 5) perceived effectiveness; 6) self-efficacy; and 7) affective attitude (S1 Table) [26, 27]. Additional questions captured pharmacist demographics and measured perceptions of the implementation plan and usability of the technology. The survey questions utilized a 5-point Likert scale ranging from strongly disagree to strongly agree or from very difficult to very easy. The survey was translated into Vietnamese (S2 Survey) and the pilot's coordinator (PMTT) verified the translation through field testing. To test the reliability of the survey instrument, we calculated Cronbach's alpha for internal consistency (S2 Table).

We calculated a sample size of 100 pharmacists, in two equal sized cohorts of 50, based on a 2-sample, 2-sided comparison of mean scores on a 5-point Likert scale, with a 95% confidence level and 80% power. We estimated an average rating difference between pharmacists of 0.9 and a population variance of 5 based on initial results from piloting the survey instrument.

Pharmacists who made at least one CXR referral which resulted in the participant getting a CXR screen via the pilot (n = 50), as well as pharmacists who agreed to participate, but did not make any CXR referrals (n = 50), were randomly approached for recruitment until the sample size requirements were met. The survey was administered in person using the open-source ONA data collection platform (https://ona.io/) on Android tablets.

## Statistical analysis

We reported pharmacy recruitment, yields along the TB care cascade and the number needed to screen (NNS) by intervention district. We divided the acceptability survey data into pharmacists who made at least one CXR referral resulting in the participant being screened by CXR via the pilot versus pharmacists who did not make any CXR referrals, and compared survey participation rates using Chi-squared tests to measure significant differences in characteristics between the two cohorts. We reported mean and median responses of the acceptability survey for each cohort and fitted mixed-effect ordinal proportional odds models to explore the association of individual acceptability factors as the primary outcome and a recorded referral as primary exposures. The model was [attitude scale measure] ~ p[made a referral] + [other covariates] + error. Age, sex, professional experience and other demographic covariates were secondary exposures and district was the random effect. In the event of small or empty cells in individual groups, we aggregated individual 5-point Likert scale responses into three categories to stabilize the model: 1) Disagree, 2) Neither disagree nor agree, and 3) Agree. We removed the random effect, if the inclusion thereof rendered the models unstable. Point estimates of modeled results included a 95% confidence interval. Hypothesis tests were two-sided and a threshold of p = 0.05 was considered statistically significant. Statistical analyses were performed using Stata version 14.

## Ethical considerations

Ethical approvals were granted by the Ethics Committee for Biomedical Research of the Ha Noi School of Public Health (324/2019/YTCC-HD3) and the Ethic Review Committee of the WHO Regional Office for the Western Pacific (2019.22.VTN.5.STB). Pharmacy customers provided verbal consent and could drop out of the pilot at any time without affecting the care

provide by any providers involved in this pilot. All of the surveyed pharmacists were given a participant information sheet and signed an informed consent form prior to completing the acceptability survey.

## Results

### Pharmacist outreach and recruitment

A total of 1,816 push notifications were sent to active SwipeRx users across five attempts to recruit pharmacists for the referral pilot; 55 (3.0%) of these push notifications were opened within 24 hours. In parallel, a Facebook advertisement campaign encouraged people to both join the SwipeRx platform and participate in the pilot. Together, these efforts resulted in 78 SwipeRx users opting into pilot participation. Of these, just one users (1.2%) was situated in the pilot's intervention districts. However, when she was approached, she was no longer working in that district and was not eligible to be recruited.

Using lists of officially-registered private pharmacies from the District Health Authorities, pilot staff directly approached 253 people working in private pharmacies in person, resulting in 173 (68.4%) pharmacists agreeing to participate in the pilot (Table 1). Of those, 146 (84.4%) installed the SwipeRx app and completed a training on pilot procedures. In addition to regular in-person visits by project staff, push notifications reminding collaborating pharmacists to screen and make referrals were sent via the SwipeRx app five times during a two-month period in the second half of the pilot. By the end of the pilot, 54 (37.0%) pharmacists had made at least one CXR referral.

### TB referral cascade

Fifty-four pharmacists referred 182 people for CXR screening (Table 2), for an average of 3.4 referrals per pharmacist. In reality, just two pharmacists (3.7%) accounted for 31.3% of the CXR referrals, while 35 pharmacists (64.8%) made just one or two CXR referrals throughout the entire pilot period; the remaining 17 pharmacists made between three and five referrals each. Only 64 of the people referred (35.2%) ultimately received a CXR, resulting in the detection of 24 people with an abnormal CXR (37.5%) and 22 people tested on Xpert (91.7%). An additional two people with a normal CXR result were tested off algorithm. Seven people were diagnosed with bacteriologically-confirmed TB and linked to treatment, for a NNS of 26.

### Acceptability survey recruitment

We randomly approached 106 (72.6%) of participating pharmacists for recruitment in the acceptability survey. Of these, 50 were among pharmacists who made at least one CXR referral (50/54 = 92.6%) and 56 were among pharmacists who made zero referrals during the pilot period (56/92 = 60.9%). Six persons in the latter group declined to participate, so that the final sample consisted of 100 completed surveys equally split between the two cohorts (Fig 1). Among these 100 survey participants, 80.0% were female (Table 3). Most pharmacists operated

**Table 1. In-person recruitment of pharmacists in the referral pilot by district.**

| Pharmacists | All Districts | District 05 | District 08 | District 10 | Go Vap |
|---|---|---|---|---|---|
| Approached | **253** | 14 | 53 | 39 | 147 |
| Agreeing | **173 (68.4%)** | 14 (100%) | 53 (100%) | 38 (97.4%) | 68 (46.3%) |
| Trained | **146 (84.4%)** | 8 (57.1%) | 50 (94.3%) | 29 (76.3%) | 59 (86.8%) |
| Making ≥1 referral | **54 (37.0%)** | 5 (62.5%) | 12 (24.0%) | 12 (41.4%) | 25 (42.4%) |

**Table 2. TB referral cascade among pharmacy customers by district.**

| Pharmacy Customers | All Districts | District 05 | District 08 | District 10 | Go Vap |
|---|---|---|---|---|---|
| Referred for CXR | **182** | 14 | 57 | 36 | 75 |
| CXR performed | **64 (35.2%)** | 3 (21.4%) | 7 (12.3%) | 12 (33.3%) | 42 (56.0%) |
| Abnormal CXR | **24 (37.5%)** | 1 (33.3%) | 4 (57.1%) | 7 (58.3%) | 12 (28.6%) |
| Tested on Xpert | **22 (91.7%)** | 1 (100.0%) | 2 (50.0%) | 5 (71.4%) | 14 (116.7%) |
| Xpert(+) TB | **7 (31.8%)** | 0 (0%) | 0 (0%) | 3 (60.0%) | 4 (28.6%) |
| RIF-Resistant TB | **0 (0%)** | N/A | N/A | 0 (0%) | 0 (0%) |
| Started on treatment | **7 (100%)** | N/A | N/A | 3 (100%) | 4 (100%) |
| NNS | **26** | N/A | N/A | 12 | 19 |

in Go Vap (45.0%), while the proportion in District 05 was lowest (7.0%), approximately mirroring the distribution of participating pharmacists. There was a significantly greater share of pharmacists with >15 years of experience (p = 0.02) in the cohort making at least one CXR referral.

## Acceptability of TB screening and referral

There were significant differences in pharmacist responses for six factors associated with four of the seven acceptability constructs (Table 4). Pharmacists who made at least one CXR referral

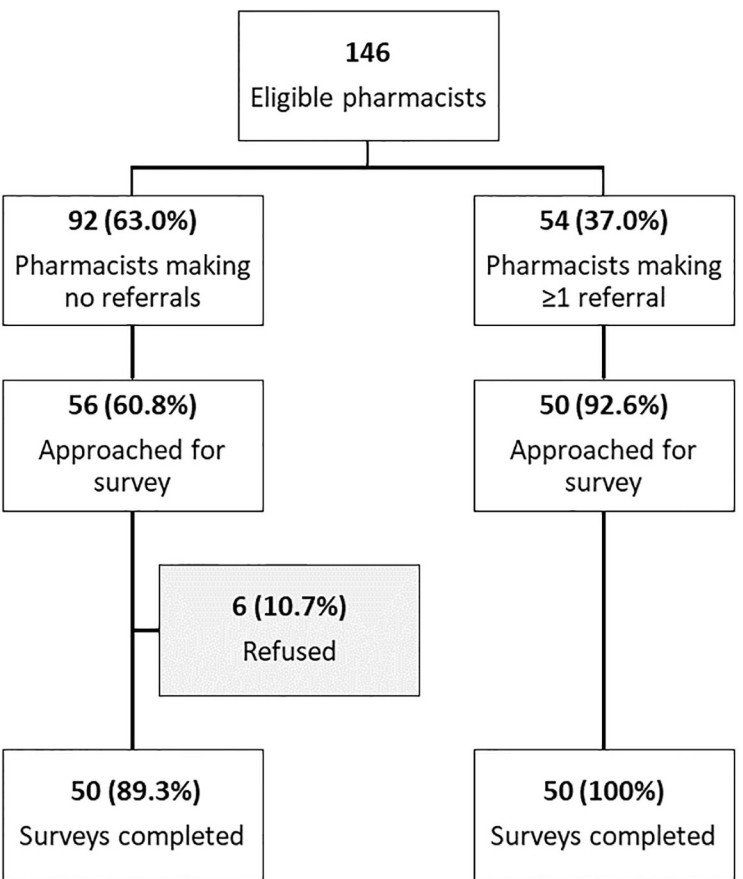

**Fig 1. Flow diagram of pharmacist recruitment for the acceptability survey.**

**Table 3. Characteristics of surveyed pharmacists, stratified by their referral status.**

| Variables | All pharmacists surveyed | Pharmacists making ≥1 CXR referral (n = 50) | Pharmacists making no CXR referrals (n = 50) | P-Value |
|---|---|---|---|---|
| **Sex** | | | | |
| Female | 80 (80.0%) | 41 (51.3%) | 39 (48.8%) | 0.67 |
| Male | 20 (20.0%) | 9 (45.0%) | 11 (55.0%) | |
| **District** | | | | |
| District 05 | 7 (7.0%) | 5 (71.4%) | 2 (28.6%) | 0.52 |
| District 08 | 27 (27.0%) | 11 (40.7%) | 16 (59.3%) | |
| District 10 | 21 (21.0%) | 11 (52.4%) | 10 (47.6%) | |
| Go Vap | 45 (45.0%) | 23 (51.1%) | 22 (48.9%) | |
| **Age (years)** | | | | |
| 18–25 years | 33 (33.0%) | 15 (45.5%) | 18 (54.5%) | 0.10 |
| 25–35 years | 29 (29.0%) | 11 (37.9%) | 18 (62.1%) | |
| >35 years | 38 (38.0%) | 24 (63.2%) | 14 (36.8%) | |
| **Years since graduation** | | | | |
| ≤5 years | 45 (45.0%) | 20 (44.4%) | 25 (55.6%) | **0.02** |
| 5–15 years | 27 (27.0%) | 10 (37.0%) | 17 (63.0%) | |
| >15 years | 28 (28.0%) | 20 (71.4%) | 8 (28.6%) | |
| **Years with private pharmacies** | | | | |
| 1–5 years | 45 (45.0%) | 20 (44.4%) | 25 (55.6%) | **0.02** |
| 5–15 years | 31 (31.0%) | 12 (38.7%) | 19 (61.3%) | |
| >15 years | 24 (24.0%) | 18 (75.0%) | 6 (25.0%) | |

were more likely to agree that the purpose of the intervention was to find TB earlier (aOR = 2.6, 95%CI: 1.2–5.8), and that they were confident in their ability to use the SwipeRx app for screening (aOR = 3.1, 95CI%: 1.4–6.8). Those who referred were also more likely to agree that the pilot would increase customer trust in their pharmacy (aOR = 2.7, 95%CI: 1.2–5.8), and that it was beneficial to his/her pharmacy (aOR = 2.8, 95%CI: 1.3–6.2). The largest difference between the two survey cohorts was in the pharmacist's belief that TB screening/referral differentiated his/her pharmacy from others in the area (aOR = 4.4, 95%CI: 1.9–9.9). There were significant differences in pharmacist perceptions of Intervention Coherence and Opportunity Cost across the four pilot districts (S4 Table). Other constructs relating to training and implementation were not associated with the probability of referrals (S3 Table). For a detailed description of responses, please see S5 Table.

## Discussion

Our results showed that novel engagement approaches, such as digital networking platforms aimed at pharmacies, can reach a broad range of providers with very little resource requirements and effort. However, while this method may be able to facilitate scaled outreach, our pilot was unable to generate substantial pharmacist engagement. Only one pharmacist responded positively in the pilot's intervention areas and was later deemed ineligible. Instead, the pilot relied on the more labor-intensive strategy of offline pharmacy engagement (i.e., 'detailing'), which managed to elicit a higher rate of agreement to participate on the pilot, and has been noted in other settings before [28, 29]. We are also able to determine key beliefs that may be associated with willingness to make a CXR referral, including the coherence of the intervention, perceived effectiveness, affective attitude, and self-efficacy. Pharmacists were able to successfully use electronic forms to collect referral data, which facilitated real-time monitoring and management of this multi-district pilot.

**Table 4. Association of individual acceptability factors with making at least one CXR referral.**

| | Pharmacists making ≥1 CXR referral (n = 50) | | Pharmacists making no CXR referrals (n = 50) | | OR (95%CI) | p-value |
|---|---|---|---|---|---|---|
| | Mean (95% CI) | Median (IQR) | Mean (95% CI) | Median (IQR) | | |
| **Ethicality[a]** | | | | | | |
| Appropriate to verbally screen | 4.0 (3.9–4.2) | 4 (4–4) | 4.0 (3.6–4.2) | 4 (4–5) | 0.9 (0.4–1.9) | 0.732 |
| Appropriate to refer without a physician | 4.1 (4.0–4.3) | 4 (4–5) | 4.1 (3.8–4.4) | 4 (4–5) | 1.4 (0.5–4.4) | 0.464*$ |
| **Intervention Coherence[a]** | | | | | | |
| Screening helps customer get an early TB diagnosis | 4.6 (4.4–4.7) | 5 (4–5) | 4.3 (4.1–4.5) | 4 (4–5) | **2.6 (1.2–5.8)** | **0.022***  |
| Screening ensures access to quality TB diagnostics | 4.0 (3.7–4.2) | 4 (3.5–4) | 3.7 (3.5–3.9) | 4 (3–4) | 2.2 (0.9–4.9) | 0.068 |
| **Opportunity Cost[a]** | | | | | | |
| Screening takes too much time | 3.0 (2.7–3.3) | 3 (2–4) | 2.7 (2.5–3.0) | 3 (2–3) | 1.6 (0.8–3.3) | 0.193 |
| Screening costs pharmacist money | 2.0 (1.9–2.1) | 2 (2–2) | 2.1 (2.0–2.1) | 2 (2–2) | 0.4 (0.1–1.6) | 0.195* |
| Screening causes customers to not return | 2.0 (1.8–2.3) | 2 (1–2) | 2.1 (1.8–2.3) | 2 (2–2) | 0.9 (0.4–2.0) | 0.84 |
| **Perceived Effectiveness[a]** | | | | | | |
| Screening helps identify more people with TB | 4.52 (4.4–4.7) | 5 (4–5) | 4.24 (4.1–4.4) | 4 (4–5) | **2.5 (1.1–5.7)** | **0.026***  |
| Screening increases customer trust in pharmacist | 4.1 (3.9–4.3) | 4 (4–5) | 3.7 (3.4–3.9) | 4 (3–4) | **2.7 (1.2–5.8)** | **0.014** |
| **Affective Attitude[a]** | | | | | | |
| Screening is beneficial to pharmacy business | 3.8 (3.6–4.0) | 4 (3–4) | 3.4 (3.2–3.6) | 3 (3–4) | **2.8 (1.3–6.2)** | **0.008***  |
| Screening differentiates my pharmacy from others | 3.7 (3.5–4.0) | 4 (3–4) | 3.1 (2.8–3.3) | 3 (2–4) | **4.4 (1.9–9.9)** | **<0.01***$ |
| **Self-Efficacy[a]** | | | | | | |
| Confidence in using ACIS/SwipeRx app to screen | 3.7 (3.5–4.0) | 4 (3–4) | 3.2 (2.9–3.5) | 3 (2–4) | **3.1 (1.4–6.8)** | **0.006** |
| Confidence in referring eligible people for CXR | 3.4 (3.2–3.6) | 3 (3–4) | 3.1(2.9–3.3) | 3 (3–4) | 2.1(1.0–4.6) | 0.055 |
| **Burden[b]** | | | | | | |
| Time requirement for customers to be screened | 3.3 (3.1–3.6) | 4 (3–4) | 3.0 (2.8–3.3) | 3 (2–4) | 1.9 (0.90–3.9) | 0.093 |
| Time requirement for customers to get a CXR | 2.9 (2.6–3.1) | 3 (2–3) | 2.8 (2.6–3.1) | 3 (2–3.5) | 1.0 (0.5–2.2) | 0.943 |

[a]: 1 (Strongly disagree) to 5 (Strongly agree)

[b]: 1 (Very difficult) to 5 (Very easy)

*: Ordinal regression model

$: In the event of small or empty cells in individual groups, we aggregated individual 5-point Likert scale responses into three categories to stabilize the model: 1) Disagree, 2) Neither disagree nor agree, and 3) Agree

Nonetheless, the acceptability survey also elucidated several key barriers to their effectiveness that translated to a "pre-screening" of customers, likely depressing the total number of referrals in our pilot. Overall, the pilot resulted in the detection and linkage to care of seven persons with TB among 182 persons referred for CXR screening. Other pharmacy engagement pilots have achieved higher CXR referral rates per pharmacist per month, while achieving similar yields of TB [15]. One key barrier in our setting was a low level of knowledge and confidence among participating pharmacists in their ability to screen for and recognize TB symptoms. A lack of knowledge or clinical skills has been cited as a common barrier to pharmacist involvement in public health and TB interventions [30–32]. Interestingly, more established pharmacists indicated a higher willingness to systematically screen their customers, possibly because they have had time to develop good relationships with long-standing customers.

A pharmacist's lack of confidence in making referrals could be driven by an unwillingness to give false advice, particularly about a stigmatized disease, such as TB. 17]. This could lead to a critical perception of the pharmacist's competency if their diagnostic tests results were negative. As pharmacists did not want to lose credibility (or 'face'), a pillar of Vietnamese

sociocultural norms, such a 'misdiagnosis' could ultimately have a detrimental impact on their business [33]. This is concordant with a previous pharmacist engagement pilot, which noted that even though it was generally acceptable to pharmacists to participate in a screening intervention, a common perceived risk of participation was losing customers to a competitor [5, 34]. Connected to that was a reported hesitancy to advocate too strongly for fear of inconveniencing their client [35]. This type of cultural reticence to engage in other people's affairs has also been noted elsewhere [36].

The pilot's low rate of CXR screening after referral from a pharmacist highlights the challenges inherent in case finding. Studies from Viet Nam have highlighted the barriers of introducing facility-based CXR screening into diagnostic algorithms [37–40]. Other studies have shown that pharmacy clients often perceive the role of pharmacies to be limited to symptom relief and the provision of commodities, rather than the dispensation of healthcare advice [15]. Pharmacy customers may seek advice from general care providers before returning to the pharmacist and then following the proposed pathway to a CXR facility [11, 41]. This behavior has been observed on patient pathway and patient cost analyses before [42, 43]. This further supports the well understood notion that investing in the training of pharmacy staff and private providers can improve the TB care and prevention ecosystem [44].

The acceptability survey also highlighted the need for public health interventions involving private sector providers to account for their economic objectives. Pharmacists who made at least one referral indicated that it was beneficial to the business of the pharmacy. Beyond pure economics, however, pharmacists also agreed that the screening intervention could identify more customers with TB earlier and raising the perceived quality of care. This is concordant with Lönnroth et al.'s study, on which Vietnamese pharmacist identified reputational gains for providing TB care as another form of reward [17]. People with presumptive TB often prefer being evaluated with more than smear microscopy, and this pilot offered the opportunity for subsidized CXR screening and free-of-cost Xpert testing (if eligible), which is not available for people seeking care in the private sector at the time. Some of the collaborating radiography sites and treatment options were also private facilities, allowing pharmacy customers to avoid engaging with the public-sector entirely, if desired.

While the anticipated recruitment of pharmacists from SwipeRx's existing user base did not materialize in the timeline planned in our pilot, the SwipeRx platform remains a potentially useful tool for rapid dissemination of information and strengthening of pharmacists' capacity. The utility of professional networking platforms for continuing medical education has been a subject of interest for some time [45] and is similarly evidenced by the efforts of expanding SwipeRx's offering portfolio with continuing medical education (CME) courses in Viet Nam. In addition to formal capacity building, studies have noted social media to offer other key benefits such as staying connected with colleagues, networking with the wider community, sharing knowledge, benchmarking and branding, which may be further explored for enhanced pharmacy engagement [46].

Particularly regarding branding, most pharmacists that made at least one CXR referral in our pilot believed that screening would differentiate the service provided by their pharmacy compared to others and would actually increase customer trust in the pharmacist. This shows promise that a properly executed screening intervention could help bolster the participating pharmacy's reputation and credibility, and strengthen pharmacist-customer relationships. By extension, utilizing social media platforms for public health interventions, such as pharmacy engagement for TB case finding, could entail highlighting the work of 'champion referrers,' which could lead to reputation gains and a positive business impact. In light of our results and the abundance of potential alternative use cases, our pilot further highlights the need for more evidence on the optimal utilization of digital networking tools in the public health space [47,

48]. Further studies assessing the perspective of pharmacy customers and beneficiaries of the diagnostic referral pilot are also needed, to understand whether the pilot and its promotion of champion referrers would actually result in increased trust in the champion pharmacy.

One major limitation of our pilot was its small scope. Specifically, the concentrated geography, short intervention time and small sample size limit the generalizability of our results. A larger intervention area and implementation period may have generated greater engagement via the SwipeRx platform. Conversely, a greater number of participants and referrals would have likely resulted in a lower yield, particularly if including both urban and rural areas. Another limitation is that we did not collect any information about the characteristics and barriers faced by pharmacists who did not respond to the push notifications encouraging them to participate in the pilot via the SwipeRx app. Just two pharmacists accounted for nearly a third of the referrals, suggesting a high potential for bias in the data. However, this could also be evidence of the Pareto principle and that in lieu of broad engagement, a strategy built on targeted strengthening of collaborations with high-volume referrers might offer a more effective avenue for scale up of screening interventions among pharmacies as it has in other private sector engagement interventions [49, 50]. Additionally, given the pre-screening of pharmacy customers, we were not able to capture the true extent of persons needed to be evaluated to detect the seven persons with TB in this pilot.

## Conclusion

The engagement of pharmacists for TB screening and referrals shows promise in this urban, high TB burden setting, as a high yield of people with TB have the potential to be identified earlier in their disease course and screening and diagnostic referrals were highly acceptable among participating pharmacists. In order to efficiently engage pharmacies for this activity at scale, more evidence on the optimal use of online and offline channels of recruitment is needed, particularly the role of professional networking and social media in identifying and reaching eligible collaborators. Despite these unanswered questions, this pilot adds to the evidence base regarding the inclusion of pharmacies in a country's private sector engagement strategy and their strategies to end TB overall.

## Supporting information

**S1 Table. Acceptability construct definitions.**
(DOCX)

**S2 Table. Cronbach's alpha (reliability coefficient).**
(DOCX)

**S3 Table. Comparison of training and implementation constructs among private pharmacists.**
(DOCX)

**S4 Table. Comparison of acceptability constructs across pilot districts.**
(XLSX)

**S5 Table. Detailed survey responses.**
(XLSX)

**S1 Survey. Private pharmacist acceptability survey in English.**
(DOCX)

**S2 Survey. Private pharmacist acceptability survey in Vietnamese.**
(DOCX)

**S1 Data. Stata code for analysis of acceptability survey data.**
(DO)

## Acknowledgments

The authors express their sincere gratitude to the Viet Nam National Tuberculosis Control Programme, Pham Ngoc Thach Hospital and the staff working at the District Health Offices for their support and referral of potential participants. The authors also wish to thank all participating pharmacists and their customers. We further acknowledge the assistance of staff from SwipeRx. We are thankful for the support from the Clinton Health Access Initiative and TechUp for the development and unrestricted availability of the ACIS digital screening and record management platform.

## Author Contributions

**Conceptualization:** Andrew J. Codlin, Rachel J. Forse, Luan N. Q. Vo.

**Data curation:** Phuong M. T. Tran, Thu A. Dam, Andrew J. Codlin, Rachel J. Forse, Luan N. Q. Vo.

**Formal analysis:** Phuong M. T. Tran, Thu A. Dam, Andrew J. Codlin, Rachel J. Forse, Luan N. Q. Vo.

**Funding acquisition:** Andrew J. Codlin, Rachel J. Forse, Luan N. Q. Vo.

**Investigation:** Phuong M. T. Tran, Thu A. Dam, Huy B. Huynh, Andrew J. Codlin, Rachel J. Forse, Luan N. Q. Vo.

**Methodology:** Phuong M. T. Tran, Huy B. Huynh, Andrew J. Codlin, Rachel J. Forse, Luan N. Q. Vo.

**Project administration:** Phuong M. T. Tran, Thu A. Dam, Huy B. Huynh, Andrew J. Codlin, Rachel J. Forse, Ha M. T. Dang, Vinh V. Truong, Lan H. Nguyen, Hoa B. Nguyen, Nhung V. Nguyen, Thuy T. T. Dong, Giang H. Nguyen, Luan N. Q. Vo.

**Resources:** Andrew J. Codlin, Rachel J. Forse, Jacob Creswell, Farouk Meralli, Fukushi Morishita, Luan N. Q. Vo.

**Software:** Andrew J. Codlin, Rachel J. Forse, Luan N. Q. Vo.

**Supervision:** Phuong M. T. Tran, Andrew J. Codlin, Rachel J. Forse, Ha M. T. Dang, Vinh V. Truong, Lan H. Nguyen, Hoa B. Nguyen, Nhung V. Nguyen, Jacob Creswell, Farouk Meralli, Fukushi Morishita, Thuy T. T. Dong, Giang H. Nguyen, Luan N. Q. Vo.

**Validation:** Phuong M. T. Tran, Thu A. Dam, Andrew J. Codlin, Rachel J. Forse, Luan N. Q. Vo.

**Visualization:** Phuong M. T. Tran, Thu A. Dam, Andrew J. Codlin, Rachel J. Forse, Luan N. Q. Vo.

**Writing – original draft:** Phuong M. T. Tran, Thu A. Dam, Andrew J. Codlin, Rachel J. Forse, Luan N. Q. Vo.

**Writing – review & editing:** Phuong M. T. Tran, Thu A. Dam, Huy B. Huynh, Andrew J. Codlin, Rachel J. Forse, Ha M. T. Dang, Vinh V. Truong, Lan H. Nguyen, Hoa B. Nguyen,

Nhung V. Nguyen, Jacob Creswell, Farouk Meralli, Fukushi Morishita, Thuy T. T. Dong, Giang H. Nguyen, Luan N. Q. Vo.

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
