## [Decision Letter · Decision Letter 0]

28 Oct 2021

PGPH-D-21-00625

Evaluating novel engagement mechanisms, yields and acceptability of tuberculosis screening at retail pharmacies in Ho Chi Minh City, Viet Nam

Dear Dr. Codlin,

Thank you for submitting your manuscript to PLOS Global Public Health. After careful consideration, we feel that it has merit but does not fully meet PLOS Global Public Health’s publication criteria as it currently stands. Therefore, we invite you to submit a revised version of the manuscript that addresses the points raised during the review process.

We look forward to receiving your revised manuscript.

Kind regards,

Andrew D. Kerkhoff

Academic Editor

Journal Requirements:

1. Please provide additional details regarding participant consent. In the ethics statement, please ensure that you have specified whether consent was informed.

2. Please provide us with a direct link to the base layer of the map used in Figure 1 and ensure this location is also included in the figure legend. 

Please note that, because all PLOS articles are published under a CC BY license (creativecommons.org/licenses/by/4.0/), we cannot publish proprietary maps such as Google Maps, Mapquest or other copyrighted maps. If your map was obtained from a copyrighted source please amend the figure so that the base map used is from an openly available source.

Please note that only the following CC BY licences are compatible with PLOS licence: CC BY 4.0, CC BY 2.0  and CC BY 3.0, meanwhile such licences as CC BY-ND 3.0 and others are not compatible due to additional restrictions. If you are unsure whether you can use a map or not, please do reach out and we will be able to help you. 

The following websites are good examples of where you can source open access or public domain maps:

3. Please provide separate figure files in .tif or .eps format only, and remove any figures embedded in your manuscript file.  If you are using LaTeX, you do not need to remove embedded figures.

4. If you have no competing interests to declare, please state "The authors have declared that no competing interests exist".

5. In the online submission form, you indicated that your data will be submitted to a repository upon acceptance.  We strongly recommend all authors deposit their data before acceptance, as the process can be lengthy and hold up publication timelines. Please note that, though access restrictions are acceptable now, your entire data will need to be made freely accessible if your manuscript is accepted for publication. This policy applies to all data except where public deposition would breach compliance with the protocol approved by your research ethics board. If you are unable to adhere to our open data policy, please kindly revise your statement to explain your reasoning and we will seek the editor's input on an exemption. Please be assured that, once you have provided your new statement, the assessment of your exemption will not hold up the peer review process.

6. State the initials, alongside each funding source, of each author to receive each grant.

Additional Editor Comments (if provided):

In addition to the reviewer's queries and suggestions, please also address the following points.

- Abstract:

o Line 27: Recommend clearly linking in the second or third sentence that SwipeRx platform is a professional networking app. As written, it is implied but not fully clear.

o Line 37: Add a comparator for the sentence below to better contextualize results (e.g., compared to whom?).

o Line 43: For the sentence, as is written, “Digital networking platforms, such as SwipeRx, can facilitate pharmacist engagement, but…” does not seem to be supported by the results provided (e.g., at best 78/1816 ~4.3%; or 1/1816 within the intervention area ~0.6%). I would suggest that you more clearly demonstrate what supports this statement (e.g., that referrals were made through app after in-person outreach), or provide greater nuance to your conclusion in the abstract.

- Manuscript:

o Methods: For the acceptability survey, how did the author’s decide which sub-set from each group to recruit (e.g., the 50/54 and the 56/92)? Please provide clarification how you decided whom to survey from among those that participated.

o Line 139: Please reference S2 Table after noting checking internal consistency via Cronbach’s alpha.

o Table 1: Recommend somehow capturing the two different methods of outreach. As is, it’s a bit misleading since most people outreached via a push notification declined to take part.

o Line 204: The results text here mentions 6 acceptability factors across four constructs, yet the methods text mentions 7 acceptability constructs. Please reconcile these differences. Do the authors mean that 4 of 7 acceptability constructs were higher among those making at least one referral?

o Discussion/limitations: Given the very low engagement following digital outreach, were there any efforts to understand the characteristics of pharmacists invited via push notification but who did not participate (and how they compared to those who agreed) and their barriers to engagement / reasons for non-participation? Since this constituted 95% of those originally outreached to, it would be extremely informative to understand determinants related to why they didn’t accept/participate. If this data was not collected, please note this as a study limitation.

o Conclusion: Please provide greater nuance to your conclusions as currently the challenges of engagement/adoption around digital outreach are not mentioned. This is an important aspect of your findings (e.g., while adoption via digital outreach was low, among those reached, the screening intervention was highly acceptable).

Reviewers' comments:

Reviewer's Responses to Questions

**Comments to the Author**

1. Does this manuscript meet PLOS Global Public Health’s publication criteria? Is the manuscript technically sound, and do the data support the conclusions? The manuscript must describe methodologically and ethically rigorous research with conclusions that are appropriately drawn based on the data presented.

Reviewer #1: Partly

Reviewer #2: Yes

2. Has the statistical analysis been performed appropriately and rigorously?

Reviewer #1: No

Reviewer #2: Yes

3. Have the authors made all data underlying the findings in their manuscript fully available (please refer to the Data Availability Statement at the start of the manuscript PDF file)?

Reviewer #1: No

Reviewer #2: Yes

4. Is the manuscript presented in an intelligible fashion and written in standard English?

Reviewer #1: Yes

Reviewer #2: Yes

5. Review Comments to the Author

Reviewer #1: Dear Editor,

Many thanks for inviting me to review this paper by Tran et al. . This is a useful paper that shows the impact (and perhaps pitfalls) of a novel approach to engaging private sector providers to improve TB diagnosis in Ho Chi Minh City.

The first part of the paper describes the intervention that used two mHealth applications (SwipeRx and ACIS), and showed that pharmacists and/or their staff can effectively link people with TB symptoms to TB diagnosis and care. However, the authors note that engagement using an online-app-only invitation may not be sufficient as none of the pharmacist participants were recruited solely through push notifications on the app. They also note that some pharmacists made many more successful referrals than others (a third of all TB screening referrals came from two pharmacists).

The second part of the paper details pharmacists’ attitudes about the intervention and factors associated with referring a pharmacy client for chest Xray.

The intervention is interesting, and it’s great to read about new and innovative ways to improve TB diagnosis and linkage to care. I enjoyed reading the first part and the descriptive analysis, but I have some reservations about the analysis and reporting of the association between pharmacist acceptability factors and referrals.

Major comments

My major comments are around the reporting of the acceptability survey.

The Introduction states that the intention was to identify factors with a pharmacist’s active participation in the pilot, and the Methods state that recorded referral was the primary outcome and attitudes the primary exposure. However, in the results sentences are phrased as if the modelled association was the other way around – for example “those who referred were also more likely to agree that the pilot would increase customer trust…”.

I have several related comments here:

1. It’s not completely clear to me from description in methods and results including Table 3 what way around the multivariable model is set up.

I think that, as currently reported, these are probably odds ratios for a one unit increase on a five-item Likert scale to predict whether a pharmacist has made a referral or not? i.e. the left hand side of model is whether a referral was made or not, and the right hand side is the attitude score (p[made a referral] ~ [attitude score] + [other covariates] + error).

I would be grateful if the authors could provide additional details to clarify what exactly is being modelled here, and which model structures were used.

2. However, I think because of the way the pharmacists have been purposefully sampled that the exposure (i.e. attitudes) needs to be on the left hand side of the model and whether someone made a referral or not on the right hand size. Looking at the methods, it seems as if pharmacists were purposefully sampled to separately recruit equal numbers of those who did and did not refer patients. I think this is probably equivalent to a case-control study. It follows therefore that the authors should be aiming to compare exposures (i.e. acceptability measures) between pharmacists with given levels of outcome. I don’t think they can directly estimate odds ratios for outcomes (referral vs. no referral) since data were collected / pharmacists were purposefully sampled conditional on whether a pharmacist had made a referral or not, and therefore the authors already determined that half of all included pharmacists had made a referral.

Unless there is something I have missed, I think the correct model set-up would be [attitude scale measure] ~ p[made a referral] + [other co-variates] + error.

Because attitude measures are on a 5-level scale, I think this would probably best be a multinomial model. A multinomial model would avoid making assumptions of linearity (or log-linearity) between the Likert scale levels (i.e. would avoid assumption that a difference from 1 to 2, it the same as the difference from 3 to 4). The estimand here then wouldn’t be an odd ratios, but an expected difference in attitude score in those who did vs. those who did not make a referral.

3. Although it is mentioned by the authors, I wonder if there was much evidence for clustering of attitudes by District? If so, I don’t think these results (i.e. district-specify effects and variation) have been fully reported. I wonder if the authors might increase the precision of estimates if they considered responses nested within pharmacists rather than districts (for example, the pharmacists who think screening takes too much time are also likely to think screening costs pharmacists too much money). I think this would require modelling of all attitude scores at once, rather than in many separate models, but I think this could probably be handled in an item-response model. This is just a suggestion.

4. In Table 3, rather than showing mean and median attitude response, the authors should show numbers and proportion of people selecting each 1 to 5 response. If it can’t sensibly fit into a table, I think it should at least be supplementary material.

5. Understanding of this would be clearer if an equation was included in the manuscript, or the statistical code was made available (I note that PLoS Global Public Health recommends that code be made freely available).

6. For the sample size calculation, the authors write that the sample size is based on the “probability of a pharmacist making referrals when their attitude is at the mean of a five point likert scale”. But when the probability of referral is what? What probability of referral at the mean of five point scale did they use? That said, I think if the analysis substantially changes this power calculation may become moot.

Minor comments

7. I would be interested to read a little more in introduction or methods about the healthcare system set-up in Viet Nam. For instance, the paper mainly talks about “pharmacists” but sometimes about “pharmacy professionals”. Are all the people included qualified pharmacists? (vs. people who work in retail pharmacy, perhaps under supervision of an off-site pharmacist)? For instance, in Malawi, if I went to pharmacy retail outlet, the person I would speak with would be unlikely to be a trained pharmacist, although a trained pharmacist might be responsible for the retail outlet overall. What is the regulatory framework in Viet Nam? What level of medical education do these people tend to have? Similarly, in comments about private vs. public sector I’d be interested to know about costs in the public sector, are chest Xrays free? And what are costs to people for TB treatment in public vs. private sector?

This need not be long, but a sentence or two would help me understand a little more about the system in which the intervention is situated.

8. In some places the authors talk about “made a referral” vs. “made a successful referral”. What is the definition of making a “successful” referral? Does this require the referred person to actually receive a chest Xray? Or just that it went through on the mHealth app?

9. In line 114 to 122, the authors talk about people with “suspected TB”. Whilst “people with suspected TB” is infinitely better than “TB suspects”, I still think this isn’t preferred language (https://www.ghdonline.org/uploads/Zachariah_et_al_2012_Language_in_TB_services_can_we_change_to_patient-centred_terminology_and_stop_the_paradigm_of_blaming_the_patients.pdf). Would “people with presumptive TB” or “people who had been identified as having TB symptoms” be better?

10. In line 252 the authors write about what the pharmacists think their customers want from a service and line 277 about how participating in the pilot might lead to increased trust from customers. I would perhaps add to the discussion the limitation that only pharmacists were surveyed, so we don’t directly know what pharmacy customers perceive / prefer or what would actually lead to an increase in customer trust - merely what the pharmacists themselves think.

11. The authors probably don’t need both p values and confidence intervals, confidence intervals are probably sufficient on their own.

Trivial comments or typos

12. I am not sure “bifurcated” is the most usual word to use for dividing data. Would “divided according to” be better?

13. The authors refer throughout to “multivariate” models, I think they probably mean “multivariable” (although if they change the modelling approach as suggested above, this might change).

14. Line 233 to 235 seems overly long, could this be split up? Something along the lines of, “Probably pharmacists conducted ‘pre-screening’ of customers to only refer those with more severe symptoms rather than all those meeting symptom screening criteria, likely depressing the total number of referrals. Overall the intervention resulted in detection and linkage to care of seven persons with TB among 182 persons referred for CxR screening” ?

Reviewer #2: Thank you for the opportunity to review this paper. It is very well written, easy to follow and covers an important issue in the need to find ways of identifying, diagnosing and treating the many people with TB who are “hidden” or appear late for treatment. Results from this pilot will be valuable in going forward to develop a larger study engaging more pharmacists in Viet Nam.

I only have the following minor comments that would help clarify some issues.

1. It would be good to have a bit more information about pharmacies in Viet Nam and the pharmacists included in the study ie. do the pharmacists tend to work alone in their pharmacy or are they working alongside others? Is it the trained pharmacist who might be working in the background while an assistant is the main interface with customers?

2. Line 87: How were your pilot districts selected?

3. Lines 97 & 98: What is the definition of eligibility? Were push notifications only sent to pharmacists in the pilot area? Later in the results it appears this was not the case but it would be good to have this clearer in the methods

4. Line 124: What is meant by “successful CRX”. Is this that only referral was made, or referral and CXR done, or that the patient was TB positive? Suggest removing the word “successful”.

5. Line 133: Is there a reference for theoretical framework of acceptability?

6. Line 172: I would argue that your 84.4% who installed the SwipeRx rather overstates participation. This should instead be 146 of the 253 approached (58%).

7. Lines 173-175: Its seems that 207 notifications were sent to the 146 participating pharmacists which is just over 1 notification per pharmacist over the 4-month period. This doesn’t seem very much given the supposed ease of doing these notifications. Can you please clarify and/or discuss this more in the discussion.

8. Line 180 & 181: 54 pharmacists referred patients for CXR: 2 pharmacists made many referrals and 35 made just one or two. What happened to the other 27 pharmacists?

9. Lines 190-193: How did you recruit the pharmacists who made zero referrals? Was this random selection?

10. Line 244: Pharmacist feedback - “unwillingness to give false advice”. I didn’t see any reference to this in the results. Was this informal feedback or was it part of the questions in Table 4?

11. Lines 296-298: Only 35% of patients referred actually received a CXR. This seems a very small proportion yet you have left it to just these two lines to mention it. I realise your paper is more about referral but it would be good to see more discussion on this.

6. PLOS authors have the option to publish the peer review history of their article (what does this mean?). If published, this will include your full peer review and any attached files.

**Do you want your identity to be public for this peer review?** For information about this choice, including consent withdrawal, please see our Privacy Policy.

Reviewer #1: No

Reviewer #2: No

---

## [Decision Letter · Decision Letter 1]

2 Jul 2022

PGPH-D-21-00625R1

Evaluating novel engagement mechanisms, yields and acceptability of tuberculosis screening at retail pharmacies in Ho Chi Minh City, Viet Nam

Dear Dr. Codlin,

Thank you for submitting your manuscript to PLOS Global Public Health. After careful consideration, we feel that it has merit but does not fully meet PLOS Global Public Health’s publication criteria as it currently stands. Therefore, we invite you to submit a revised version of the manuscript that addresses the points raised during the review process.

We look forward to receiving your revised manuscript.

Kind regards,

Andrew D. Kerkhoff

Academic Editor

Journal Requirements:

1. Please update your online Competing Interests statement. If you have no competing interests to declare, please state: “The authors have declared that no competing interests exist.”

Additional Editor Comments (if provided):

Thank you for your revised submission. The manuscript has been substantially strengthened and we appreciate your hard work on these revisions. We believe that we will be able to accept your manuscript for publication after addressing the minor points below and the remaining query from reviewer #2.

- Abstract: Kindly structure your abstract according to journal’s formatting guidelines.

- Please ensure that the remainder of your manuscript accords with the journal’s formatting guidelines.

- Please carefully review the text of your manuscript one final time for any small grammatical errors or typos as there will not be a formal proofing stage, nor a further opportunity to revise your manuscript.

Reviewers' comments:

Reviewer's Responses to Questions

**Comments to the Author**

1. If the authors have adequately addressed your comments raised in a previous round of review and you feel that this manuscript is now acceptable for publication, you may indicate that here to bypass the “Comments to the Author” section, enter your conflict of interest statement in the “Confidential to Editor” section, and submit your "Accept" recommendation.

Reviewer #1: All comments have been addressed

Reviewer #2: All comments have been addressed

2. Does this manuscript meet PLOS Global Public Health’s publication criteria? Is the manuscript technically sound, and do the data support the conclusions? The manuscript must describe methodologically and ethically rigorous research with conclusions that are appropriately drawn based on the data presented.

Reviewer #1: (No Response)

Reviewer #2: Yes

3. Has the statistical analysis been performed appropriately and rigorously?

Reviewer #1: (No Response)

Reviewer #2: Yes

4. Have the authors made all data underlying the findings in their manuscript fully available (please refer to the Data Availability Statement at the start of the manuscript PDF file)?

Reviewer #1: (No Response)

Reviewer #2: Yes

5. Is the manuscript presented in an intelligible fashion and written in standard English?

Reviewer #1: (No Response)

Reviewer #2: Yes

6. Review Comments to the Author

Reviewer #1: Thank you for revision, all my comments have been addressed. I think that the revised statistical model looks much more sensible and the paper looks great and really clear. I particularly like all the supplementary material provided and open data approach. Whilst acknowledging that acceptance decisions are with the editor rather than reviewers, I look forward to seeing your paper in print.

Reviewer #2: Thank you for addressing my queries. I just have one outstanding thing that is not yet clear and I hope you can address. On Page 3 you say “Most retail pharmacies in Viet Nam employ staff with minimal formal pharmacy training, who then work under the supervision of an off-site pharmacist”. Then on Page 5 you say “We engaged any staff member working at the pharmacy who directly dispensed and consulted clients in the pilot, regardless of their official certification or degree”.

Did your definition of pharmacist and hence the recruitment include only the “off-site pharmacist” or did it also include those with no official training? It seems that it is the latter but then Table 3 makes it seem like it was only trained pharmacists. It would be good to clarify this and also the implications of this in the discussion.

Thank you

7. PLOS authors have the option to publish the peer review history of their article (what does this mean?). If published, this will include your full peer review and any attached files.

**Do you want your identity to be public for this peer review?** For information about this choice, including consent withdrawal, please see our Privacy Policy.

Reviewer #1: No

Reviewer #2: No

---

## [Editor Report · Decision Letter 2]

29 Aug 2022

Evaluating novel engagement mechanisms, yields and acceptability of tuberculosis screening at retail pharmacies in Ho Chi Minh City, Viet Nam

PGPH-D-21-00625R2

Dear Mr Codlin,

We are pleased to inform you that your manuscript 'Evaluating novel engagement mechanisms, yields and acceptability of tuberculosis screening at retail pharmacies in Ho Chi Minh City, Viet Nam' has been provisionally accepted for publication in PLOS Global Public Health.

Best regards,

Andrew D. Kerkhoff

Academic Editor